

# Horizontally transferred genes in the ctenophore *Mnemiopsis leidyi*

Alexandra M. Hernandez[1,2] and Joseph F. Ryan[1,2]

[1] Whitney Laboratory for Marine Bioscience, St. Augustine, FL, USA
[2] Department of Biology, University of Florida, Gainesville, FL, USA

## ABSTRACT

Horizontal gene transfer (HGT) has had major impacts on the biology of a wide range of organisms from antibiotic resistance in bacteria to adaptations to herbivory in arthropods. A growing body of literature shows that HGT between non-animals and animals is more commonplace than previously thought. In this study, we present a thorough investigation of HGT in the ctenophore *Mnemiopsis leidyi*. We applied tests of phylogenetic incongruence to identify nine genes that were likely transferred horizontally early in ctenophore evolution from bacteria and non-metazoan eukaryotes. All but one of these HGTs (an uncharacterized protein) are homologous to characterized enzymes, supporting previous observations that genes encoding enzymes are more likely to be retained after HGT events. We found that the majority of these nine horizontally transferred genes were expressed during development, suggesting that they are active and play a role in the biology of *M. leidyi*. This is the first report of HGT in ctenophores, and contributes to an ever-growing literature on the prevalence of genetic information flowing between non-animals and animals.

## INTRODUCTION

Evolution is commonly thought to occur by descent with modification from a single lineage. However, evidence has shown that genomes from bacteria, archaea, and eukaryotes are typically chimeric, resulting from horizontal (or lateral) gene transfers (*Garcia-Vallvé, Romeu & Palau, 2000*; *Katz, 2002*). As such, horizontal gene transfer (HGT) has likely impacted evolution more than originally thought by creating opportunities for rapid genetic diversification and contributing to speciation events. Moreover, HGT is a potential catalyst for organisms to acquire novel traits (*Soucy, Huang & Gogarten, 2015*) and creates opportunities for HGT receivers to exploit new ecological niches (*Boto, 2010*). For example, HGTs have played an important role in herbivory in arthropods (*Wybouw et al., 2016*), venom recruitment in parasitoid wasps (*Martinson et al., 2016*), cellulose production in urochordates (*Dehal et al., 2002*) and plant parasitism in nematodes (*Haegeman, Jones & Danchin, 2011*).

Although HGT is generally accepted as an important evolutionary mechanism in prokaryotes (*Boto, 2014*), it remains controversial whether it occurs in animals, despite many convincing studies (*Madhusoodanan, 2015*). Much of the skepticism has been fueled

Corresponding author
Joseph F. Ryan,
joseph.ryan@whitney.ufl.edu

by high-profile reports of HGT (*Lander et al., 2001*; *Boothby et al., 2015*) that were later shown to be largely incorrect due to contamination or taxon sampling (*Stanhope et al., 2001*; *Koutsovoulos et al., 2016*). In addition, HGT in animals is hypothesized to be rare due to the origin of a sequestered germ line, which provides fewer opportunities for germ cells to be exposed to foreign DNA (*Doolittle, 1999*; *Andersson, Doolittle & Nesbø, 2001*; *Jensen et al., 2016*). However, the presence and absence of germline sequestration is not well described across the animal tree of life, and there are inconsistencies between studies regarding which animal groups have sequestered germlines (*Buss, 1983*; *Radzvilavicius et al., 2016*; *Jensen et al., 2016*).

The major challenges for HGT detection efforts have been taxon sampling and contamination. Many early reports of HGT in animals were overturned due to limited representation of taxa in public genomic databases (*Salzberg et al., 2001*). For example, a gene present in bacteria and humans, but absent from nematodes and drosophilids (the most highly represented taxa at the time) may have been considered the result of HGT, until discovering that the gene is present in many other animal genomes that were not available at the time of the initial claim. In these cases, the limited representation of taxa made it difficult to distinguish HGTs from differential gene loss (*Andersson et al., 2006*; *Keeling & Palmer, 2008*). More recently, contamination has led to both overestimation and likely underestimation of HGT events. In several recent cases, contamination in newly generated datasets has been interpreted as HGT but later shown to be cross-contaminants present in genome sequences (*Bhattacharya et al., 2013*; *Delmont & Eren, 2016*; *Koutsovoulos et al., 2016*). On the other hand, the presence of contaminants in public databases (e.g., a bacteria sequence labeled as an animal sequence) makes it difficult to identify *bona fide* HGTs, as "animal" sequences will appear among the top BLAST hits for a particular HGT, leading to false negatives (*Kryukov & Imanishi, 2016*). As such, contamination remains a major hurdle to contemporary studies of HGT.

Pairwise BLAST-based similarity scores (e.g., alien index (*Gladyshev, Meselson & Arkhipova, 2008*) and the HGT index (*Boschetti et al., 2012*)) are the most common criteria used to detect HGT in animals. However, these measures largely ignore phylogenetic information associated with sequence data. While a positive BLAST-based result may be due to HGT, it may also result from gene loss, selective evolutionary rates, convergent evolution, sequence contamination, and species misassignment (*Hall, Brachat & Dietrich, 2005*). Previous HGT studies have demonstrated that HGT predictions need to be carefully considered and a combination of methods are required to rule out false positives (*Schönknecht, Weber & Lercher, 2014*). Hypothesis tests incorporating phylogenetic incongruence are one such method that has been used to test HGT. While some studies in animals have incorporated these techniques (*Eliáš et al., 2016*), they are more commonly deployed in studies involving non-animals (*Bapteste, Moreira & Philippe, 2003*; *Richards et al., 2006*).

HGT has yet to be thoroughly explored in Ctenophora. Ctenophores (comb jellies) are marine invertebrates that are morphologically characterized by eight rows of cilia used for movement. They typically live in the water column, but the group includes benthic species as well (*Song & Hwang, 2010*; *Alamaru, Brokovich & Loya, 2015*; *Glynn et al., 2017*).

Phylogenomic evidence from studies including ctenophores has suggested that ctenophores are the sister group to all other animals (*Dunn et al., 2008*; *Hejnol et al., 2009*; *Ryan et al., 2013*; *Moroz et al., 2014*; *Borowiec et al., 2015*; *Chang et al., 2015*; *Torruella et al., 2015*; *Whelan et al., 2015*; *Arcila et al., 2017*; *Shen, Hittinger & Rokas, 2017*; *Whelan et al., 2017*), but the position remains controversial with some evidence supporting sponges as the sister group to the rest of animals (*Philippe et al., 2009*; *Pick et al., 2010*; *Pisani et al., 2015*; *Telford, Budd & Philippe, 2015*; *Simion et al., 2017*; *Feuda et al., 2017*). Thus, investigating HGT in ctenophores is essential to understanding its implications on early animal evolution.

Here, we apply a rigorous framework to identify and confirm HGTs in the ctenophore *Mnemiopsis leidyi*. Our process includes identification of HGT candidates by alien index and confirmation by phylogenetic hypothesis testing to provide statistical support in an evolutionary framework. Furthermore, we analyze gene expression profiles during development to obtain clues as to the function of these HGTs in *M. leidyi*.

## MATERIALS AND METHODS

### Identification of HGT candidates by alien_index

As part of this project, we developed the program alien_index and complimentary metazoan/non-metazoan sequence databases to automate the generation of alien index (*Gladyshev, Meselson & Arkhipova, 2008*) and HGT index scores (*Boschetti et al., 2012*). We BLASTed the entire set of *M. leidyi* gene models (ML2.2) (*Ryan et al., 2013*) against a database of animal and non-animal sequences (alien_index_db version 0.01) and then calculated alien index values as the logarithmic difference between the best BLASTP *E*-values for animal and non-animal hits (as outlined in *Gladyshev, Meselson & Arkhipova (2008)*) (Fig. 1A). In more simple terminology, the alien index reflects the difference between the *E*-value of the best non-animal BLAST hit and that of the best animal hit. The database used includes translated gene models from curated genomes that include bacteria (5), archaea (2), non-animal eukaryotes (5), and animals (12). See Table S1 or http://ryanlab.whitney.ufl.edu/downloads/alien_index/ for the entire list of taxa. HGT index values were computed by the difference in the highest non-animal and animal bit scores generated from the alien_index database. The alien_index program is available at: https://github.com/josephryan/alien_index.

### Confirmation of HGTs

We applied a phylogenetic approach to confirm putative HGTs. HGT candidates from alien_index were used as queries for BLASTP against NCBI's RefSeq database (*O'Leary et al., 2016*) using the NCBI BLAST interface. We collected the top 10 sequences each from bacteria, eukaryotes, fungi, and animals with an *E*-value cutoff of 0.1. We included only the first sequence if there were hits to sequences from species in the same genus (Fig. 1B). We also added the top BLAST hit (*E*-value ≤ 0.1) from each of the following fully sequenced animals from version 0.01 of the alien_index database: *Amphimedon queenslandica*, *Trichoplax adhaerens*, *Nematostella vectensis*, *Capitella teleta*, *Drosophila melanogaster*, and *Homo sapiens*. Sequences were aligned against the corresponding

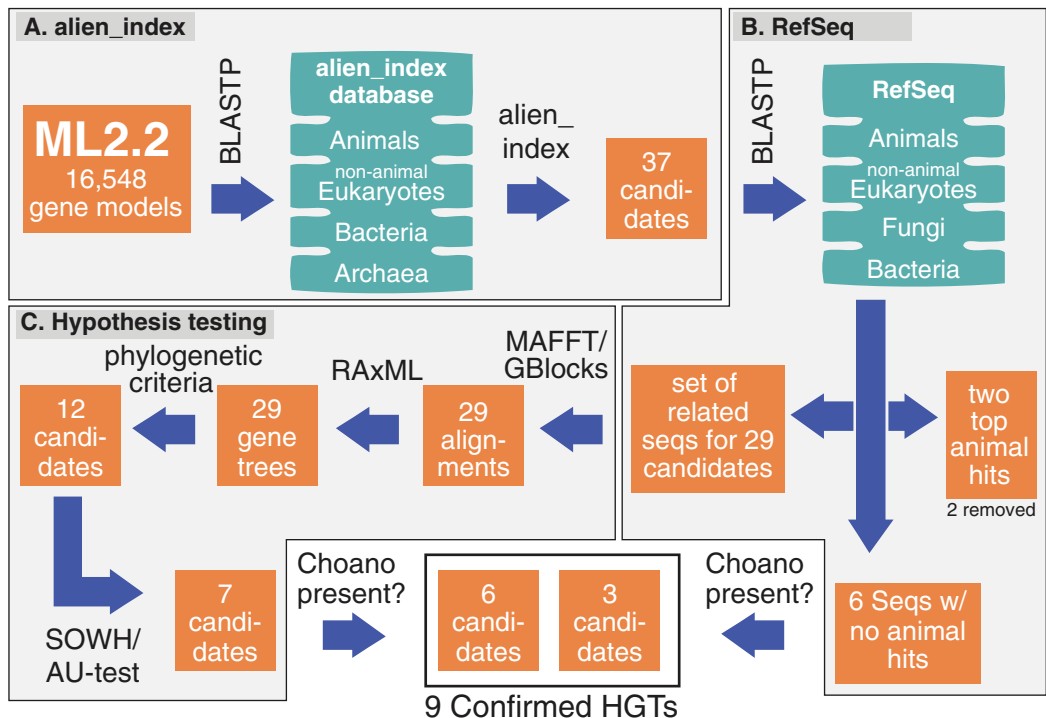

**Figure 1 Pipeline and outputs to identify and confirm HGTs.** (A) alien_index was used to identify 37 HGT candidates. (B) These candidates were then BLASTed against RefSeq; two candidates were removed because they only had two significant animals hits and six were set aside for future testing because they lacked animal hits. (C) The remaining 29 candidates were tested by phylogenetic analyses and hypothesis testing (SOWH and AU test). The six candidates that lacked animal hits and those that passed hypothesis testing (seven candidates) were screened for significant hits to Choanoflagellates. More details on genes passing through the pipeline are described in Table S2.     

putative HGT using MAFFT (*Katoh et al., 2002*; *Katoh & Standley, 2013*) and trimmed with Gblockswrapper (*Castresana, 2000*) (Fig. 1C). There were six genes (ML012034a, ML06718a, ML03277a, ML02232a, ML18354a, ML219316a) with BLASTP hits to non-animals but not to animals (*E*-value $\leq$ 0.1), preventing us from performing additional phylogenetic analyses on these sequences. We considered the lack of animal BLASTP hits below our cutoff as sufficient evidence that these six were clearly HGTs. ML018031a and ML00882a only had two BLASTP hits to animal sequences. Since it was unclear if this resulted from contamination, we were unable to test these genes using phylogenetic approaches, so they were removed from contention as HGTs.

We performed maximum-likelihood analyses on the remaining 29 alignments using RAxML (*Stamatakis, 2014*) (Fig. 1C). Since the RefSeq database has many instances of contamination (*Pible et al., 2014*), we allowed a maximum of two non-ctenophore animal sequences to fall outside of the main animal clade. To implement this, we pruned putative contaminants if the removal of two taxa resulted in a monophyletic animal clade (Fig. S1). We discarded any HGT candidates with more than two taxa disrupting animal monophyly.

We explicitly tested topologies in opposition to HGT (i.e., animal monophyly) with the SOWH test using SOWHAT (*Church, Ryan & Dunn, 2015*) and the AU test using CONSEL
(*Shimodaira & Hasegawa, 2001*) (Fig. 1C). The SOWH and AU test evaluate statistical support for phylogenetic incongruence by comparing the likelihood values between trees to a distribution of trees generated by parametric sampling in the SOWH test and non-parametric sampling in the AU test. We required that these two different approaches to hypothesis testing agreed to ensure that our criteria confirming *bona fide* HGTs was stringent. To address any potential problems of selection bias in the AU test (causing the likelihood value to bias upwards for the maximum likelihood best tree when included in the dataset), we performed multiple AU analyses using bootstrap trees as suboptimal trees (similar to *Eliáš et al., 2016*). We generated 100 bootstrap trees using RAxML rapid bootstrap analyses, and verified there were no duplicate trees in our 100 bootstrap set using the ape package in R (*Paradis, Claude & Strimmer, 2004*). RAxML was used to generate per-site log likelihoods for the best maximum-likelihood tree, the tree constraining the putative HGT to metazoans (i.e., metazoan-constraint tree), and suboptimal trees, to perform the AU test implemented through CONSEL. To test the effectiveness of comparing to bootstrap trees, we manually created a set of suboptimal trees for each HGT candidate by shuffling clades of three (Fig. S2) and performed the same analyses. We evaluated the tree space covered by suboptimal trees in the AU test (i.e., bootstrap and manually generated trees) by visualizing the data using violin plots. We calculated likelihood proportions for each tree by dividing individual likelihood scores by the average likelihood score of suboptimal trees. This was done to make the data comparable for visualization since the likelihood scores differ between sets of gene trees. The trees and scripts used to automate these phylogenetic analyses are available in the accompanying GitHub site (https://github.com/josephryan/2018-Hernandez_and_Ryan_HGT).

We verified that HGT candidates were not the result of bacterial contaminants by using the *M. leidyi* genome browser (*Moreland et al., 2014*) to examine the intron/exon structure of each HGT candidate, as well as the origin of their neighboring genes. We examined each intron to determine whether it was actively handled by spliceosomes (which are only found in eukaryotes), since bacteria, archaea, and viruses contain Group I and II introns. U2 spliceosomal introns were identified by conserved GT dinucleotides at the 5' end and conserved AG dinucleotides at the 3' end of introns. We also conducted reciprocal best BLASTP searches for each HGT candidate (identified from the genome and gene models from *M. leidyi* individuals collected in Woods Hole, MA, USA) against the transcriptome of an *M. leidyi* individual collected from St. Augustine, FL, USA, as well as seven other ctenophore transcriptomes reported in *Moroz et al. (2014)*: *Bolinopsis infundibulum, Beroe abyssicola, Dryodora glandiformis, Pleurobrachia bachei, V. multiformis, Coeloplana astericola, Euplokamis dunlapae.* For these searches we used default parameters and an *E*-value cutoff of 0.1.

## HGT developmental expression profiles

An extensive transcriptomic developmental timecourse of *M. leidyi* was recently generated from single-embryos over the first 20 h of embryogenesis (*Levin et al., 2016*). To examine whether HGTs might play a role in development we used these data (GSE70185), as well as additional time points for *M. leidyi* generated after this publication (GSE111748).

The *Levin et al. (2016)* data was produced by using three replicate timecourses that each consisted of 20 isolated embryos from fertilization to 20 h. Embryos were flash frozen and RNA was extracted with TRIzol and sequenced using Illumina sequencing according to the CEL-Seq protocol (*Hashimshony et al., 2012*). For each replicate, reads were mapped to *M. leidyi* gene models (ML2.2) using bowtie2 version 2.2.3 (*Langmead & Salzberg, 2012*) with default settings and reads per transcript were counted using htseq-count (*Anders, Pyl & Huber, 2015*). Normalization of read counts was performed by dividing by the total number of counted reads and multiplying by $10^6$. Since the CEL-Seq protocol involves sequencing only from the 3' end of transcripts, results are not normalized by length of transcript.

Since the publication of *Levin et al. (2016)*, six additional time points (four replicates each) for hours 14–19 (not included in the original study) have been sequenced and submitted to the Gene Expression Omnibus (GSE111748). These additional data were produced by the same researchers (i.e., Itai Yanai and Mark Martindale) from the original study using the same methods and facilities. To create a baseline for what is considered adequate expression during development, we summed median transcripts per million (tpm) values for all replicates along the 25 time points for each of our nine confirmed HGTs. HGTs that had summed median read counts of 100 or greater were classified as being expressed sufficiently to have roles in development. We chose a value that was 10 times stricter than the minimum criteria in *Levin et al. (2016)* (i.e., 10 transcripts) to err on the side of caution.

## HGT origins and functions

To uncover the functional roles of HGTs, we used the BLAST interface provided by UniProt and the UniProtKB database (*Pundir, Martin & O'Donovan, 2017*) to identify homologous sequences used to characterize genes. Annotations of the top hits (*E*-value $\leq 0.1$) were assigned to HGT candidates. We also associated HGTs with Pfam-A domains using the MGP Portal under the *Mnemiopsis* Gene Wiki (*Moreland et al., 2014*). In all cases, the annotations based on BLAST and Pfam-A analyses were consistent with the results from our phylogenetic analyses. To identify the origin of the HGTs lacking animal hits (ML012034a, ML18354a, ML219316a), we performed phylogenetic analyses on the sequences collected at the start of the study from RefSeq using RAxML.

## RESULTS

### *Mnemiopsis leidyi* HGTs

Figure 1 shows our pipeline and results for each method during this analysis. We calculated an alien index for every *M. leidyi* gene model using a database of 12 animals and 12 non-animals (Table S1). We identified 37 genes with alien indices greater than 45 and designated these as HGT candidates (Fig. 1A; Table S2). In addition to the alien_index database, we BLASTed the RefSeq database at NCBI restricting hits to bacteria, then to animals, and then to non-animal eukaryotes (Fig. 1B). All but six HGT candidates had BLAST hits to animals with *E*-values $\leq 0.1$. We classified

these six (ML012034a, ML06718a, ML03277a, ML02232a, ML18354a, ML219316a) as absent from all other animals. We analyze these six further below.

For the remaining 29 candidates, we conducted detailed phylogenetic analyses using the top 10 hits of unique non-animal and animal taxa from each of the RefSeq searches along with sequences from *A. queenslandica*, *T. adhaerens*, *N. vectensis*, *C. teleta*, *D. melanogaster*, and *H. sapiens* that were top hits from our initial BLASTs of the alien_index database (Fig. 1C). HGT candidates that formed a clade with all other animals were ruled out as potential HGTs, while candidates that disrupted animal monophyly were tested further. We discarded 14 candidates with more than two non-ctenophore animal sequences disrupting animal monophyly; in cases of two or less sequences, the disrupting sequences were considered potential contaminants and pruned (e.g., Fig. S1). We discarded three more candidates after pruning because the trees continued to result in a non-monophyletic clade of animals. We then applied the SOWH and AU tests to the remaining 12 candidates to compare the maximum-likelihood topology to the alternative hypothesis that HGT candidates were more closely related to animals (Fig. 2). This involved comparing likelihood values of optimal trees to those that were constrained to produce a monophyletic Animalia. Our results showed that the AU test was more conservative in confirming HGTs than the SOWH test (Table 1). For perspective on how optimal trees compared to constrained trees, we performed AU tests comparing optimal and constrained trees to bootstrap trees (Fig. 3). The likelihood scores of the constrained trees from HGTs supported by the AU test tend to fall outside or on the tails of the distribution of likelihood scores of suboptimal trees, whereas the likelihood scores of constrained trees for unsupported HGTs were all closer to the most likely tree than the bootstrap trees (Fig. 3). We confirmed seven HGTs in which gene trees significantly differed ($p < 0.05$) from the metazoan-constraint trees in both the SOWH and AU analyses (Table 1).

We then analyzed the BLAST results of the seven HGT candidates confirmed by phylogenetic analyses and the six genes which were absent in other animals (ML012034a, ML06718a, ML03277a, ML02232a, ML18354a, ML219316a). We removed four of these genes from contention (ML092610a, ML06718a, ML03277a, ML02232a) because the top BLAST hits against RefSeq were either Choanoflagellatea or Filasterea (two of the closest protistan lineages to animals (*Hehenberger et al., 2017*; *Torruella et al., 2015*)) (*E*-value $\leq 0.1$). If ctenophores are the sister group to the rest of animals, vertical inheritance remains a possibility for these cases. As such, these tests support a total of nine HGTs.

We used the *M. leidyi* genome browser to examine the intron/exon structure of each of these nine HGTs, as well as the origin of their neighboring genes for evidence of bacterial contamination (lack of introns would indicate bacterial contamination). Eight HGTs were found on scaffolds with intron-containing genes and eight HGTs contained introns (Table 2). ML49231a (itself containing six introns) is the only gene on its scaffold. All intron-containing genes had U2 spliceosomal introns (except ML219315a, a gene on the same scaffold as an HGT). These data suggest that the majority of the HGT candidates did not appear to be bacterial contaminants.

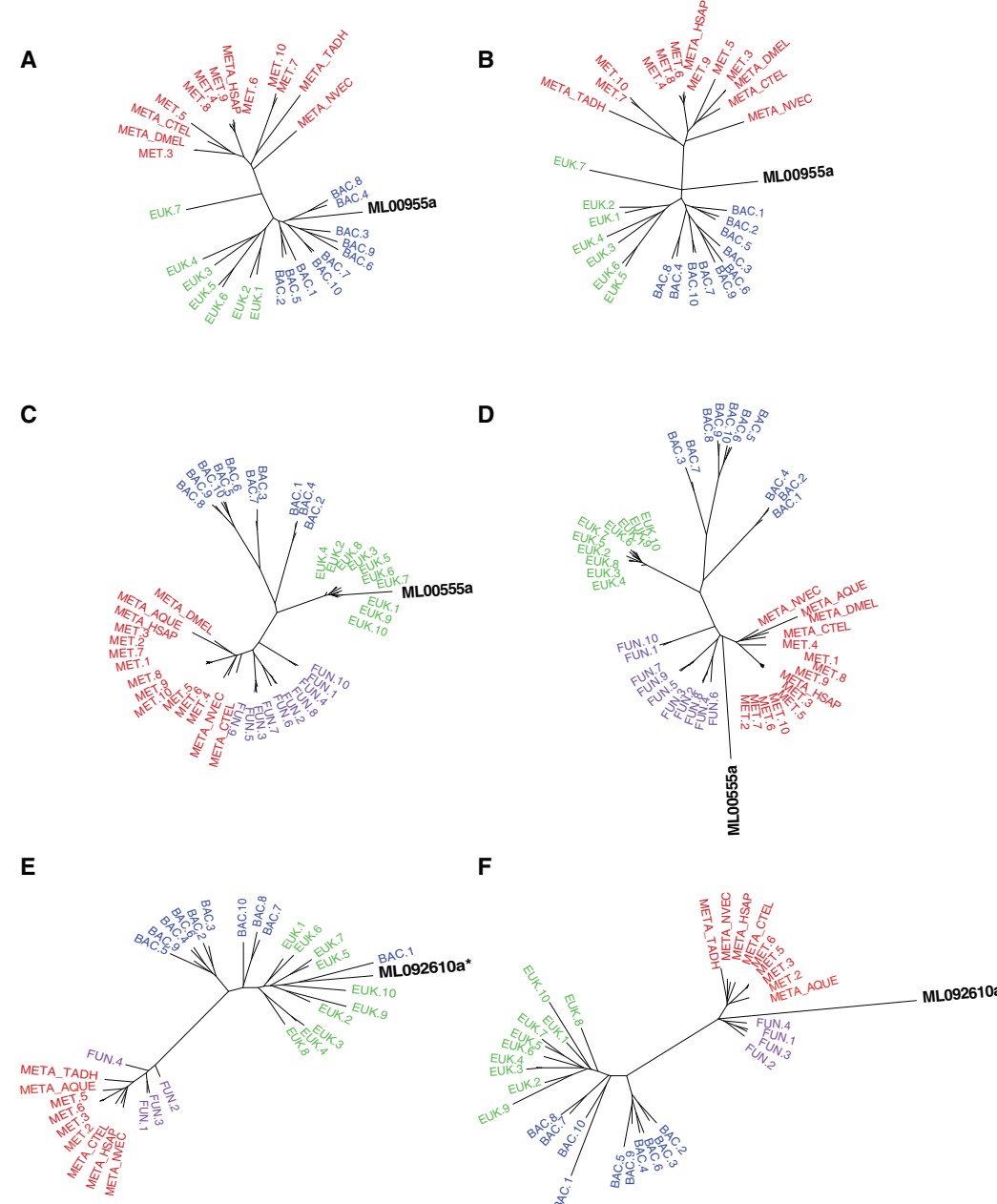

**Figure 2 Maximum-likelihood best tree and metazoan-constraint tree compared in the SOWH and AU tests.** Gene IDs (in black) denote the putative HGTs. (A), (C), and (E) are examples of RAxML best trees for HGT candidates validated by phylogenetic analyses and hypothesis testing. (B), (D), and (F) are examples of trees where putative HGTs have been constrained to produce monophyletic Animalia and have been optimized in RAxML. Taxa that are prefixed "META_" are from our alien_index database version 0.01 (i.e., META_NVEC (*Nematostella vectensis*), META_TADH (*Trichoplax adhaerens*), META_HSAP (*Homo sapiens*), META_CTEL (*Capitella teleta*), META_DMEL (*Drosophila melanogaster*), META_AQUE (*Amphimedon queenslandica*). MET = Metazoa; BAC = Bacteria; EUK = Eukaryota; FUN = Fungi; More details for each taxon are specified in Table S3. The asterisk indicates a gene that is later removed from contention.

**Table 1 Hypothesis testing on HGT candidates that were confirmed by phylogenetic analyses.**

| Genes | SOWH $p$-value | AU Bootstrap $p$-value | AU Manual $p$-value |
|---|---|---|---|
| ML00555a | <0.001 | 4.00E-45 | 7.00E-06 |
| ML49231a | <0.001 | 2.00E-44 | 7.00E-103 |
| ML092610a* | <0.001 | 2.00E-31 | 4.00E-68 |
| ML005129a | <0.001 | 1.00E-04 | 6.00E-06 |
| ML00955a | <0.001 | 0.021 | 0.002 |
| ML02771a | <0.001 | 0.023 | 0.029 |
| ML42441a | <0.001 | 0.047 | 0.022 |
| ML177319a | <0.001 | 0.226 | 0.042 |
| ML120721a | <0.001 | 0.48 | 0.245 |
| ML049014a | 0.985 | 0.862 | 0.604 |
| ML070218a | 0.262 | 0.849 | 0.361 |
| ML102910a | 0.229 | 0.719 | 0.255 |

Note:

$p$-values indicate the level of support for HGTs in comparison to the metazoan-constraint tree for the SOWH test and suboptimal trees (bootstrap and manually generated) in the AU test. Candidates in blue have significant values in all three tests ($p \leq 0.05$). The asterisk indicates a gene that is later removed from contention.

To further test these nine genes for contamination, we confirmed using BLAST that the genes were also present in a transcriptome from an *M. leidyi* individual collected from St. Augustine, FL (*M. leidyi* genome and gene models were from individuals collected in Woods Hole, MA). We also performed reciprocal best BLASTP searches for each of the nine HGTs against seven of the ctenophore transcriptomes published in *Moroz et al. (2014)*: *B. infundibulum, B. abyssicola, D. glandiformis, P. bachei, V. multiformis, C. astericola, E. dunlapae.* Each HGT was present in the transcriptome of at least one other ctenophore species and in the Florida *M. leidyi* transcriptome (Fig. 4). Furthermore, the gene lacking introns (ML012034a) is expressed in all examined ctenophore transcriptomes. Because it is unlikely that the same species contaminated each of these datasets, these comparisons provide additional evidence against HGT sequences resulting from contamination. Here, we have included as much evidence as possible to carefully confirm nine HGTs in *M. leidyi*.

## HGTs are expressed in development

We summed tpm values (medians for each set of expression values at 25 time points) from single-embryo RNA-Seq analyses over 20 h for each of the nine confirmed HGTs to identify those HGTs that were expressed during development. Six of the nine HGTs had sums greater than 100 (Fig. 5), suggesting that these had some role in development. ML00955a was expressed maternally (0 h post fertilization (hpf)) and throughout early cleavage stages (1–3 hpf) with reduced expression later in development. Three genes (ML005129a, ML18354a, ML012034a) were expressed later in development with spikes during tentacle morphogenesis (9–12 hpf). ML02771a and ML219316a displayed cyclic expression throughout development suggesting a potential role in cell cycle.

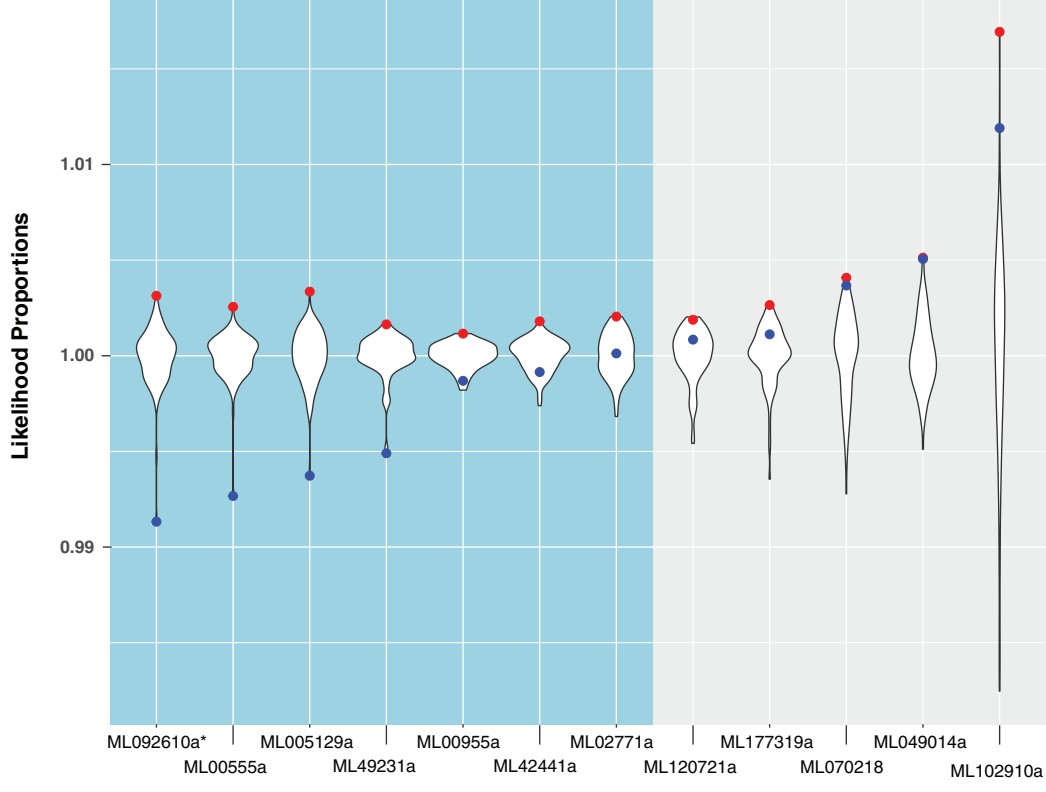

**Figure 3** **A comparison of likelihood proportions between the best tree, metazoan-constrained tree, and bootstrap trees for HGT candidates with BLAST hits to Metazoa.** Likelihood proportions are individual likelihood values divided by the average likelihood value for suboptimal trees (i.e., bootstrap trees). Red points indicate likelihood proportions of the best tree (i.e., tree indicating HGT). Blue points indicate likelihood proportions of the metazoan-constrained tree (i.e., tree contradicting HGT). The violin plot shows the distribution of likelihood proportions of 100 bootstrap trees for each HGT candidate. The side in teal shows HGT candidates validated by hypothesis testing and the side in gray shows HGT candidates unsupported by hypothesis testing. The asterisk indicates a gene that is later removed from contention.

## HGTs are enzymes originating from non-animal eukaryotes and bacteria

We used phylogenetic evidence to determine the origin of these nine HGTs. Four HGTs originated from bacteria and five from non-animal eukaryotes (Table 3). We found no evidence of HGTs that were transferred from Archaea. Specific lineage origins of three HGTs appear to be from Proteobacteria, Firmicutes, and Rhodophyta. We were unable to identify the lineage origins of the remaining HGTs. To characterize gene function, we BLASTed the nine confirmed HGTs against the UniProt database. All HGTs except one uncharacterized protein (ML219316a) were homologous to known characterized enzymes (Table 3).

## DISCUSSION

### HGTs in ctenophores and their implications

It had been previously speculated that ctenophores had HGTs since initial profiling revealed that many "bacteria-like" genes in ctenophores contained introns and were on

**Table 2  Intron structure of nine HGTs and surrounding genes.**

| Candidate HGT | Number of introns in candidate HGTs and surrounding genes | | | | | | |
|---|---|---|---|---|---|---|---|
| **ML00955a** | ML00952a | ML00953a | ML00954a | ML00955a | ML00956a | ML00957a | ML00958a |
| | 5 | 0 | 4 | 1 | 0 | 7 | 2 |
| **ML18354a** | ML18351a | ML18352a | ML18353a | ML18354a | ML18355a | ML18356a | ML18357a |
| | 6 | 14 | 7 | 5 | 1 | 0 | 0 |
| **ML02771a** | – | – | – | ML02771a | ML02772a | ML02773a | ML02774a |
| | – | – | – | 7 | 8 | 16 | 3 |
| **ML012034a** | ML012031a | ML012032a | ML012033a | ML012034a | ML012035a | ML012036a | – |
| | 6 | 5 | 6 | 0 | 3 | 6 | |
| **ML005129a** | ML005126a | ML005127a | ML005128a | ML005129a | ML005130a | ML005131a | ML005132a |
| | 0 | 7 | 2 | 1 | 0 | 4 | 8 |
| **ML219316a** | ML219313a | ML219314a | ML219315a | ML219316a | ML219317a | – | – |
| | 6 | 3 | 7* | 4 | 6 | – | – |
| **ML00555a** | ML00552a | ML00553a | ML00554a | ML00555a | ML00556a | ML00557a | ML00558a |
| | 0 | 12 | 0 | 14 | 3 | 0 | 5 |
| **ML42441a** | – | – | – | ML42441a | ML42442a | ML42443a | ML42444a |
| | – | – | – | 1 | 14 | 3 | 10 |
| **ML49231a** | – | – | – | ML49231a | – | – | – |
| | – | – | – | 6 | – | – | – |

**Note:**
The genes highlighted in red are the HGT candidates. The gene with an asterisk indicates one non-spliceosomal intron.

chromosomes with vertically inherited (i.e., non-HGT) genes (*Artamonova et al., 2015*). We confirmed that all HGTs except ML012034a had spliceosomal introns and were on scaffolds with other spliceosomal intron-containing genes (Table 2). This provided evidence that these candidates were not the result of extrinsic contamination. We provided additional evidence that candidates were not contaminants by showing that all HGTs were found in both Massachusetts and Florida *M. leidyi* individuals, as well as in many other ctenophore species (Fig. 4). Six HGTs were present in the *E. dunlapae* transcriptome suggesting that the majority of these HGT events occurred very early in ctenophore evolution (Fig. 4). This deep evolutionary history suggests that these HGTs may have had important impacts on the biology of ctenophores.

## Mechanisms driving HGT in ctenophores

While we are uncertain about the mechanisms driving HGT, we speculate that some of these may have resulted from symbiotic relationships with bacteria and non-animal eukaryotes. *Proteobacteria* is the most abundant group of bacteria associated with ctenophores (*Daniels & Breitbart, 2012*) and have been identified as donors of the gene ML00955a in the *M. leidyi* genome (Table 3) and confirmed in almost all other ctenophore transcriptomes (Fig. 4). Other possible donors could have been gymnamoebae symbionts that have been described living on the surface of comb plates and on the ectoderm of ctenophores (*Moss et al., 2001*). However, studies investigating symbiotic relationships with ctenophores are limited. Future studies are needed to improve our understanding

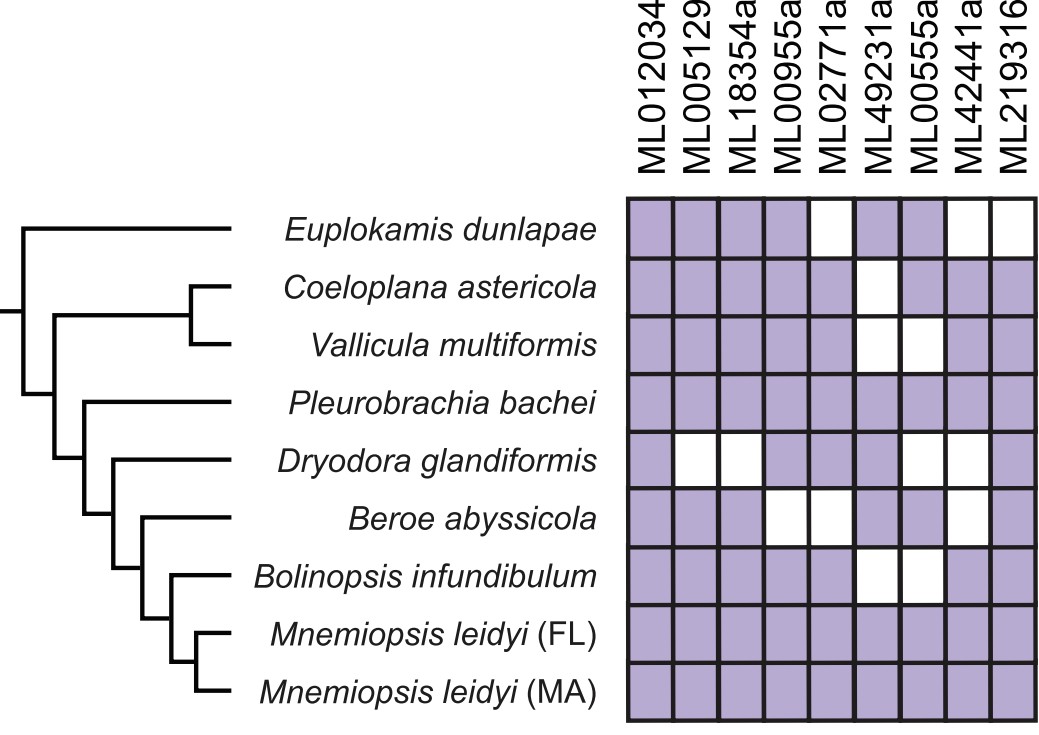

**Figure 4 Expression of confirmed HGTs from the *M. leidyi* genome in ctenophore transcriptomes.** Purple boxes indicate the specified HGT is present in the species' transcriptome confirmed by reciprocal best BLAST hits; white boxes indicate the gene is absent in the species' transcriptome. Tree was inferred by *Moroz et al. (2014)*. Percent identity among genes are described in Table S4.

of how symbiotic relationships impact HGT, as well as to understand the mechanisms that drive HGT between organisms.

## *Mnemiopsis leidyi* HGTs are expressed during development and encode enzymes

Many HGTs are likely to be deleterious and lost, but some HGTs will be neutral or provide a selective advantage and spread throughout a population (*Thomas & Nielsen, 2005*). HGT integration is thought to mainly occur in neutral genes with low levels of expression (*Park & Zhang, 2012*). Once integrated, neutral HGTs may become a source of novel genetic variation upon which selection can act (*Soucy, Huang & Gogarten, 2015*). HGTs may then become more highly expressed after recruitment of transcription factors and regulators from the host genome (*Lercher & Pál, 2008*). Six of the nine HGTs we identified showed high expression during the first 20 h of development, suggesting potentially important developmental roles (Fig. 5). ML02771a is expressed during development and encodes penicillin acylase or amidase, which catalyzes the hydrolysis of benzylpenicillin. This reaction creates key intermediates for penicillin synthesis and may be important to defend against microbial infection or colonization.

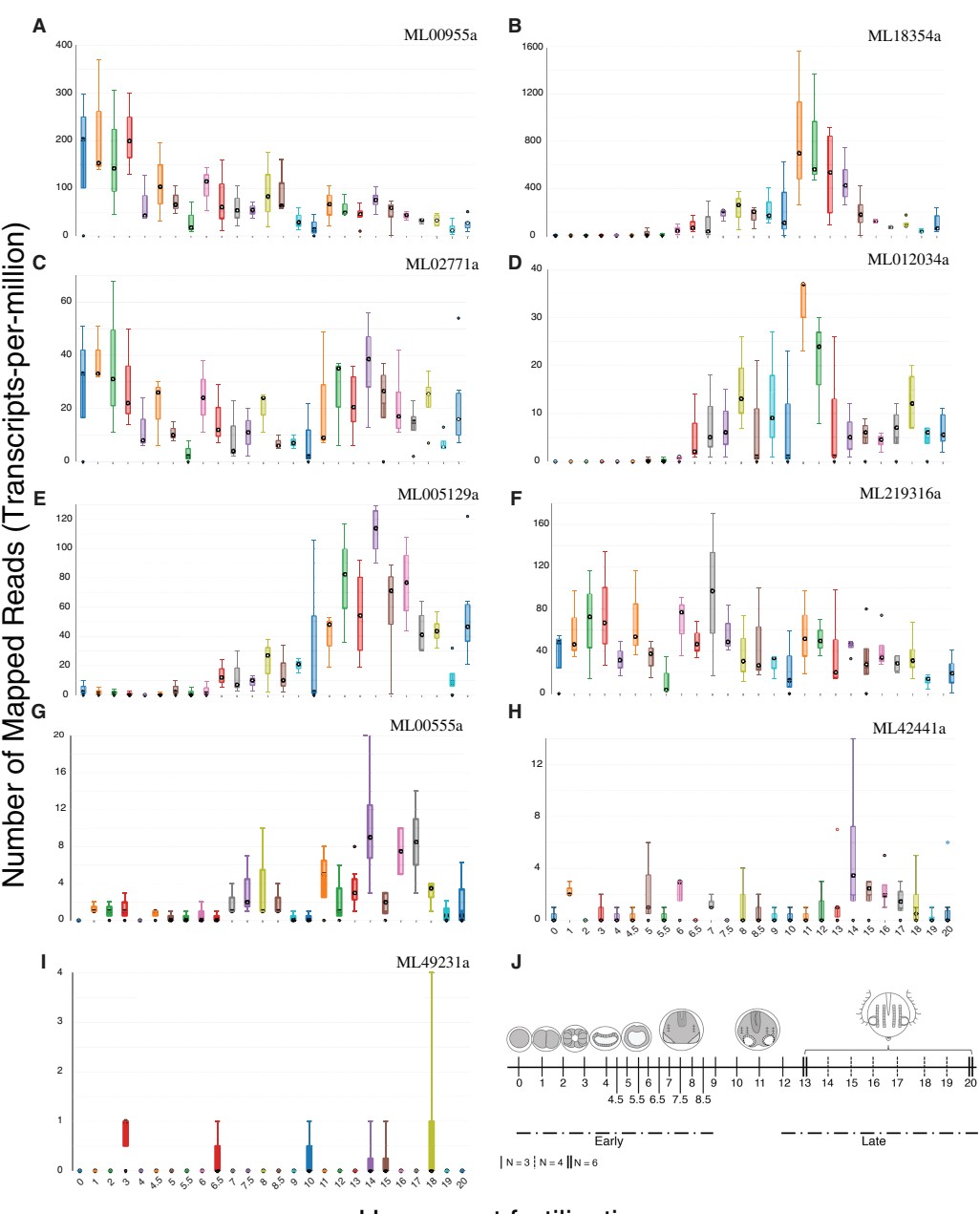

**Figure 5 Expression profiles of the nine HGTs identified in this study.** Single-embryo RNA-Seq analyses were performed over 20 h. (A–F) Confirmed HGTs with tpm values (medians for each set of time point replicates) greater than or equal to 100 over 20 h (25 time points) are shown. (G–I) Confirmed HGTs with tpm values less than 100 over 20 h. (J) Ctenophore stages of development over the timecourse. Early cleavage stages occur at 1–3 hpf. Gastrulation occurs at 4–6 hpf. Tentacle morphogenesis occurs at 9–12 hpf. *N* refers to the number of replicates.

Observations of HGT patterns in prokaryotes have also suggested that there is a preference to retain operational genes (e.g., metabolic enzymes) rather than informational genes (*Lawrence & Roth, 1996*; *Jain, Rivera & Lake, 1999*; *Garcia-Vallvé, Romeu & Palau, 2000*).

**Table 3 Sumary of confirmed HGT origins and functions.**

| Genes | Function | Pfam domains | Origin | Lineage |
|---|---|---|---|---|
| ML00955a | Putative metalloendopeptidase | Peptidase family M13 | Bac | Proteobacteria |
| ML005129a | 2-oxoglutarate (2OG) and Fe(II)-dependent oxygenase superfamily protein | 2OG-Fe(II) oxygenase superfamily | Euk | Unknown |
| ML02771a | Penicillin acylase | Penicillin amidase | Bac | Unknown |
| ML012034a | Uncharacterized protein | 2OG-Fe(II) oxygenase superfamily | Euk | Unknown |
| ML18354a | Putative chalcone and stilbene synthase | Chalcone and stilbene synthases, 3-Oxoacyl- synthase III, FAE1/Type III polyketide synthase | Bac | Unknown |
| ML219316a | Uncharacterized protein | – | Bac | Firmicutes |
| ML00555a | Phospholipase D alpha 1 | C2, Phospholipase D | Euk | Unknown |
| ML49231a | Phospholipase D gamma 1 | Phospholipase D | Euk | Rhodophyta |
| ML42441a | NADH dehydrogenase, putative | Pyridine nucleotide-disulphide oxidoreductase | Euk | Unknown |

**Note:**
HGT functions were determined by BLAST against the UniProt database and associated Pfam-A domains were searched on the *Mnemiopsis* Genome Portal. The origin column shows the domains of life from which these genes are predicted to have been transferred (Bac = Bacteria; Euk = Eukaryota). The RefSeq column shows a more detailed classification for the origin of HGTs. All rows highlighted in orange indicate genes that show developmental expression.

Informational genes, such as those involved in DNA replication, transcription, and translation are seldom found in sets of HGTs (*Thomas & Nielsen, 2005*). This propensity for operational genes is thought to occur because informational genes are involved in larger and complex systems (*Jain, Rivera & Lake, 1999*). Recently, this pattern has also been observed in animal HGTs (*Boto, 2014*) (*Zhu et al., 2011*; *Boschetti et al., 2012*; *Sun et al., 2013*; *Eyres et al., 2015*; *Conaco et al., 2016*). These reports suggest that operational genes are preferentially transferred and/or retained in both prokaryotes and eukaryotes. Our data support this idea since all of the characterizable genes in our HGT set encode enzymes.

## Commonly used BLAST-based methods for identifying HGTs in animals are insufficient

Identifying HGTs can be challenging due to bacterial associations with hosts (*Artamonova & Mushegian, 2013*; *Chapman et al., 2010*; *Fraune & Bosch, 2007*), DNA extraction kits and reagents that have led to contamination (*Naccache et al., 2013*; *Salter et al., 2014*), and/or laboratory conditions that can contaminate preparations during DNA extraction (*Laurence, Hatzis & Brash, 2014*; *Strong et al., 2014*). These challenges associated with sequencing and assembly have resulted in contamination in public databases (*Longo et al., 2011*; *Merchant, Wood & Salzberg, 2014*) and make HGT predictions difficult. Moreover, while BLAST-based approaches (i.e., alien index and the HGT index) are useful for identification of HGT candidates, they are difficult to implement, lack an evolutionary perspective, and do not address problems associated with contamination.

To overcome some of these challenges, we developed alien_index to automate the generation of alien index and HGT index scores for rapid identification of HGT candidates. We confirmed HGTs by using rigorous phylogenetic approaches to address the

problems associated with the lack of evolutionary perspective from BLAST methods. Our phylogenetic tests of incongruence provided clear metrics from which to judge the level of certainty applied to each HGT candidate. Our study showed that many of the predictions based on BLAST did not stand up to hypothesis testing and suggest that the similarity between sequences that cause high alien indices do not necessarily provide true phylogenetic signal. Consequently, incorporation of phylogenetic likelihood-based methods are necessary when performing HGT analyses in animals.

## CONCLUSION

The importance of HGT as an evolutionary mechanism in prokaryotes and eukaryotes has been underestimated. While studies of HGT in animals are gradually becoming more accepted, many challenges remain to quantify the extent of HGT and its impacts. To mitigate some of these challenges, rigorous approaches that employ both BLAST- and phylogenetic likelihood-based methods should be applied to future HGT studies in animals. Here we provided evidence of nine cases of HGT in ctenophores by applying these rigorous methods (among others), and found similar patterns of transfer between prokaryotes and eukaryotes with preference for operational genes. It should be noted that we implemented an extremely conservative approach and there are likely to be more HGTs in *M. leidyi*. However, many more studies will be necessary to gain a comprehensive overview of HGT and the mechanisms by which HGT occurs in animals.

## ACKNOWLEDGEMENTS

We would like to thank Melissa DeBiasse for constructive comments on earlier versions of this manuscript. We would also like to thank Guifré Torruella, Anthony Moss, and one anonymous reviewer whose comments and suggestions significantly improved this paper.

### Funding

This work was supported by startup funds to Joseph F. Ryan from the University of Florida DSP Research Strategic Initiatives and the Office of the Provost. This material is based upon work supported by the National Science Foundation under Grant No. (1542597). Alexandra M. Hernandez received support from the McKnight Doctoral Fellowship Program. The funders had no role in study design, data collection and analysis, decision to publish, or preparation of the manuscript.

### Grant Disclosures

The following grant information was disclosed by the authors:
University of Florida DSP Research Strategic Initiatives and the Office of the Provost.
National Science Foundation: 1542597.
McKnight Doctoral Fellowship Program.

### Competing Interests

The authors declare that they have no competing interests.

## Author Contributions

- Alexandra M. Hernandez performed the experiments, analyzed the data, prepared figures and/or tables, authored or reviewed drafts of the paper, approved the final draft.
- Joseph F. Ryan conceived and designed the experiments, contributed reagents/ materials/analysis tools, prepared figures and/or tables, authored or reviewed drafts of the paper, approved the final draft.

## Data Availability

GitHub: https://github.com/josephryan/2018-Hernandez_and_Ryan_HGT.

Gene Expression Omnibus (GEO) Database: GSE111748.

## Supplemental Information

Supplemental information for this article can be found online at http://dx.doi.org/ 10.7717/peerj.5067#supplemental-information.

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
