# Peer review of "Horizontally transferred genes in the ctenophore Mnemiopsis leidyi"

_PeerJ, doi:10.7717/peerj.5067_

## Round 0.1 · original submission · Minor Revisions

Greetings authors,

I have now read your manuscript and received 3 positive and helpful reviews. I agree with the reviewers that your manuscript is of high quality and importance. Although each reviewer has raised relatively few issues, the cumulative amount of corrections you will have to make is a bit large. On the other hand, the majority of proposed corrections and clarifications are straight-forward.

Please address all issues that the reviewers have pointed out.

Importantly, I suggest that you pay close attention to the following issues, as I believe clarifying them will strengthen your manuscript:

1. Methodology - it seems all three reviewers had trouble following distinct parts of your methodology. Please try to clarify as much as possible what has been pointed out, including tables in the main text if necessary.

2. Please clarify the caveats in conclusions and title about expression, as criticized by reviewer #3.

3. Please take in consideration the analytical suggestions for increasing taxon sampling as suggested by reviewer #1. Reviewer #3 made a similar comment about taxon sampling for the BLAST database. If these critiques are not granted, please clarify why.

·

Basic reporting

English and structure are ok (in particular, raw data, including scripts and command lines are all already freely available on-line), and the manuscript represents a clear unit of publication.

Literature should be improved in order to give the reader proper and fair explanation about the phylogenetic position hypotheses in deep animal phylogeny. Also, few sentence rephrasing or clarification are needed (please see the pdf annotated file).

Experimental design

Methods are really well thought and applied for automatic detection of HGTs, including multiple approaches (BLAST, phylogeny) and criteria (intron/exon structure, neighboring genes in scaffolds, multiple ctenophore data sources) to filter possible false positives.

The main concern I have is that alignments and phylogenies for the 9 bona fide HGTs could have been done with a much broader taxon sampling to gain resolution. For example, if most ctenophores have the homolog as Mnemiopsis, why not including them? Also, adding more bacteria and eukaryotes could improve the topology and statistical support. In fact, I'd like to see the topologies in supplementary material (not a lot of work to do), and the raw newick/nexus files for the best trees. To be fair, it would be great to have contrasting topology using bayesian inference. The reason of all this is that ctenophore sequences are long-branch and extra care should be taken.

Regarding the expression analysis, even if I'm not an expert on the matter, I don't think that the extra replicates can be compared directly with the other data due to methodological biases (not real replicates from 14-19?). I would suggest authors explicit more details instead of just citing a previous study, to facilitate reader comprehension. Finally, some concerns about the results are in the annotated pdf.

Validity of the findings

'no comment'

Additional comments

This manuscript aims to find specific horitzontal gene transfer at the onset of ctenophors by using rigorous methods of sequence similarity and phelogenetics, including expression data for Mnemiopsis leidyi. The manuscript is suitable for publication after few minor revisions.

·

Basic reporting

The reporting is clear, with very few and very minor grammatical issues.
line 40; not 'less' but 'fewer'
line 106 should state Woods Hole not Woodshole
line 116 should include: "the following fully sequenced organisms: " after 'from'
line 173 should use 'after the method of' not 'by'
see line 276. I am not sure how to correct it.
line 286: ...that can contaminate preparations during...

The authors should carefully examine their references.
Hontzeas and Willerslev et al are not in the body of the manuscript
Wybouw et al. 2016, Haegeman et al. 2011, Eyres et al. 2015; Conaco et al. 2016 are not in the references
There may be others; these are just ones I happened to pick up because they looked out of place.

In the Lit Cited, there are minor formatting issues; see:
Boto
Hejnol
Moroz

Experimental design

The manuscript meets high quality of scientific design. The concept of alien index is well established. There are areas that I am uncertain of; this could be because of my limitations with the methods chosen.

in line 104, the authors state that they examined neighboring genes to the putative HGTs; I may have missed this, but how did they do this?

in line 119, I am not sure to what the authors refer when they say 'the above criteria' - could they please clarify?

When testing between the SOWHAT and nonparametic methods, what are they seeking?

line 144: How was CONSEL used specifically?

Validity of the findings

I am only concerned about what appears to be the primary test for HGTs: intron structure.

line 176: Group I and Group II introns are indeed seen in bacteria, archaea and viruses: (see Mobile DNA 5:8, J Bact 187:5437, Nuc. Acids Res22:2532). Perhaps the authors need to more closely examine to determine whether the introns they see are all actively handled by splicosomes since those are only eukaryotic. How precisely to do this is not clear to this reviewer, but there may be a method.

I am not a professional molecular statistician so many of the arguments, which I can follow technically but which certainly have implication beyond my broader experience, may answer this type of question.

Otherwise the work appears to be mathematically solid and the entire question and conclusion well stated and properly connected together.

Additional comments

At 157 it was not made clear why the additional times were taken, but the figures showed that there were important events occurring during that time period that used these specific putative HGTs. What are these events? what sorts of tissue/cellular reorganization are occurring? this is to this reader the most exciting area of your findings.

Were putative Archaean HGTs discovered at all? looks like not...I am surprised that they are not mentioned. Perhaps I have missed something in the paper.

I have included all comments so that they can be seen by the authors.

Reviewer 3 ·

Basic reporting

The manuscript shows perfect and fluent English writing, with technical descriptors well formulated and inserted in the text. The literature cited help to understand the context of the problem, horizontal gene transfer (HGT) detection, and their application in new genome studies. References to other HGTs studies and their transcription activity would be useful (see general comments). Article structure, figures and tables are well formatted and almost ready to go online. I would suggest the implementation of one table (see general comments). Also, I like to thanks the authors for having all necessary materials ( i.e. programs and databases available in github) to reproduce the results and even apply the methodology to other cases. I think it is the way to go in today’s science communications in order to present relevant results.

Experimental design

The question and challenges of HGT in new sequenced species was well defined and their implementation in the ctenophore M. leidyi was executed nicely, combining homology search methods along with phylogeny reconstruction and confirmation. The methodology explained, with excellent detail, would be useful for future analysis and detection of HGT in non-model organism.
Although I find incomplete execution and lack of details at the developmental expression profile methods and results interpretation. I understand this part in an implementation of previous research by Levin et al. 2016, where they look (among other things) into gene expression levels at the developmental transition at embryo level in different species representing different phyla, but I don’t fully understand their inclusion in this study and the way it has been addressed. First, I would like to see more details explaining data collection (RNA extraction, mapping reads, analysis, etc), along with a brief description of the embryology of the species (M. leidyi). Why look into early stages, first 20 hours, and not some adult stage too?; so you could compare transcription values (early vs. late). If someone wants to point some gene/s are expressed at a specific state, I think first they should compared the designated estate with “other” state, and second, I think it should be usefull the use of some control candidate (housekeeping gene?) in order to normalize all transcriptional environment. I understand that the manuscript just imply their HGTs “are expressed during early development”, and not “more expressed”; though still it is rather confusing not to have something to compare with.
Please, let me put this into another perspective: if instead of early stage development, we only have RNA-seq data from late state, adults from their final live state, and a few HTGs show transcription at them, then the title would change to “expressed during late development”. This statement belongs to the title, no just a sentence in the discussion trying to bring some hypothesis for future testing. I think it is great these HGTs showed transcripts, but I think we should take this with more descriptors, analysis and comparisons, and maybe formulating other hypothesis and/or points of view. Otherwise, maybe a better formulation would be: HGTs showed transcription at the stages were RNA-seq data was available (early stages).

Validity of the findings

The authors have investigated horizontal gene transfer in the ctenophore M. leidyi, and although it’s not at a massive scale in its genome, the authors have validated each of one and have proposed a exhaustive method to validate their findings, by implementing BLAST-based scores for detection of HGTs and hypothesis test with phylogeny analysis. In general, the conclusions are well stated in the horizontal gene transfer investigation.
I think the robustness and validity of transcriptional data are more than fair for today’s standards, though some extra steps should be taken care of (reference points, further analysis explanation). But I kindly disagree with their conclusions and their link to the original question, horizontal transfer detection and validation. Under my perspective, I address horizontal transfer as the main question, since it fulfill the mayor part of the manuscript, where HGTs expression fill up 3 and a half lines in the Results section.
At the same time, I disagree with the inclusion of the term “enzyme”, specially in the title; the authors described genes and their transcript levels, and yes, they find they have homology with other proteins (UniProt database), with a function more or less characterize.

Additional comments

In the manuscript submitted “Horizontally transferred genes in the ctenophore Mnemiopsis leidyi encode enzymes and are expressed during early development” the authors described a methodology to identify horizontal gene transfer events in animals, and they develop it in the unexplored Ctenophora. They have found and validate several candidates by different means in the sequenced genome of M. leidyi. Their approach could be easily translated to other new sequenced species of metazoans, which could benefice the scientist community to discern contaminants from potential HGT candidates. All the procedure is quite well outline in the manuscript (text and supplemental material), along with external material (scripts and databases).
I think it would make a worthwhile contribution to PeerJ, but I have a few suggestion (as pointed in Experimental design and Validity of the findings), along with some other points.

Moderately important points:
Title: I think including in the title the statements of “expressed during early development” and “encode enzymes” is going too far given the nature of the analysis and discussion performed. It might mislead the reader initially with the scope of the article (see Experimental design and Validity of the findings). I think a good example can be found in the paper Hespeels et al. 2015 “Against all odds: trehalose-6-phosphate synthase and trehalase genes in the bdelloid rotifer Adineta vaga were acquired by horizontal gene transfer and are upregulated during desiccation”. The authors here dissect apart a specific group of horizontal transferred genes, where it is stated not only the RNA-seq screenings but a deeper description with comparison with other genes, implementing a metabolic context with their results, and just hypothesizing that these GENE COPIES (not enzymes) might be functional.

Figures and tables were easily followed in the text. But it was quite hard for me to keep track of the candidate genes in each of the steps. I think the reader would appreciate having a table like Table S2 in which it is described which genes are candidates and which one is confirmed, but including like which are phylogenetically confirmed (7 out of 13), the ones that have no animal Blast (6 of them), the removed genes (4), highlight the ones plotted in Fig1….along with the intron presence (maybe include numer of introns?).

Using a custom database to automate the generation of alien index. I understand the use of custom db in order to minimize the inclusion of contaminants which has been wrongfully assassinated. But I think the custom db could have more curated species, why is it not included plants, green or red algae?

Minor suggestions
Line 17. Please, replace “appear to perform enzymatic activities” with “transcriptional active”.
Line 90. Author should include what is the query (M. leidyi gene models) used in this section, and maybe describe it a little; how many genes etc…
Line 93. Authors used alien_index_db version 0.01, but at the github repository it is listed version 0.02 (January 21, 2016). Why don’t use the latest one? Is there any big differences in number of sequences/taxa?
Line 97. The database included 11 animals, but in table S1 there are 12. Please, clarify.
Line 105. Confirmation of HGTs in 7 other ctenophores. What’s the query and parameters used in that search? How many M. l. genes?
Line 110. Why is it not included the species included in the family Mertensiidae (Moroz et al 2014)?
Lines 122-126. Could the authors specify what is the E-value with animal sequences for those genes; how much (%) is the query cover (HSP).
Line 155. It should be mentioned something about the Levin experiment, like we take the 3 replicas (GEO accession number ###). And then, the additional time points (include GEO ##). It would help the comprehension of this analysis.
Line 156. It should be explained the methods used in these replicas; material (flash frozen?) , RNA extraction, sequencing, etc. The methods outlines in Levin et al. 2016 are several. It should benefit the comprehension of the text and clarification of the results obtained.
Line 159. Authors used 100 or more reads to classified a gene as expressed; please explain why choose this value. As I mentioned above, it’s hard to say without a referernce point. Question: are any of these HGT genes assigned as dynamic expressed genes (as in Levin et al 2016)?
Line 172. I would like to see Table S1 with Alien Index values for each gene too.
Line 179. Would agree the authors with me that representing GC (%) content in genes and scaffolds could also validate these findings? And if so, would it be useful to state their value in the table?
Line 186. So Fig 1 has 9 genes (HGT candidates). But at this point is hard to figure out why are only represented 9, which is resolved later in the text. An implementation of table S2 is highly recommended with maybe some text explanation at this point. Also, I think Figure 1 could have more information, like instead of only colored squared, indicate the % identity among them. I am really curious about the % identity (or divergence) between the Wood Hole and the Florida individuals.
Line 217. What Blast E-values gave the 4 genes removed?
Line 231. Please rephrase like” all HGTs except one showed homology to known characterized enzymes”.
Line 238. 73% (X of 37 candidates).
Line 259. ….”disproportionately enzymes”? Sorry, I don’t understand the term.
Lines 268-271. Speculation is welcome, but saying that ML02771a is highly expressed I think is not accurate. Maybe just say “it is expressed”.
Line 281. No, genes are not enzymes. Please, rephrase.
Line 290. “, lack evolutionary perspective” I think there is something missing in the text to make sense. Please check.
Figure 2. Please check the assignment of trees in the text, (F) is showed twice.
Figure 4. The Y axis should have all the same scale for comparisons. I wonder how are the profiles of the three genes not included (but included in table 2); maybe it would make a better example of the claims assured for RNA-seq values in these stages for HGTs.

---

## Round 0.2 · Minor Revisions

Thank you for the re-submission of your interesting manuscript. The article was re-reviewed by two of the original reviewers. While they both indicate that the manuscript has been substantially improved, they also point out a few minor suggestions that I would prefer to see addressed:

1. Reviewer #1 indicates issues with interpretation and inclusion of protistan transcriptomes that are important and need to be addressed.

2. Reviewer #2 would still like to see some more clarity on methodology and figures, and I agree with the reviewer.

I am confident that these modifications will be straight-forward to include, and will improve the breadth of your work, eventually leading to a greater readership.

Thank you,

·

Basic reporting

no comment

Experimental design

no comment

Validity of the findings

no comment

Additional comments

First, thanks for all the corrections and explanations that, in my opinion, have substantially improved your manuscript. Congratulations for your work. I've marked the recommendation as minor revisions for a couple of small points (see below), but a part from that, it could be accepted as it is.

Sorry to insist but, in lines 427-428, I'd like you to cite a recent non-animal holozoan phylogeny (you already cited one but there are more recent papers, whatever fits you) and say "Choanoflagellatea or Ichthyosporea (two of the closests protistan lineages to animals)". The reason is that another protistan lineage is closer to animals than Ichthyosporea, the Filasterea (although ncbi database may not include it, you must use state-of-the-art taxonomy, and I mean since more than 10 years ago; for further information contact the multicellgenome lab in Barcelona).

Also, please be aware that taxonomic sampling could have been wider for protistan lineages (several genomes and transcriptomes not present in ncbi but publicly available) which might challenge your results, such in line 485-486, where the Amoebozoan origin could be simply an ancestral gene from both ctenophore and amoebozoan lineages that was certainly lost in fungi, choanoflagellates, etc., but still present in closely related species not comprised in your database.

Reviewer 3 ·

Basic reporting

Thank you for the opportunity to rereview the manuscript by Hernandez and Ryan entitled “Horizontally transferred genes in the ctenophore Mnemiopsis leidyi”. The authors have extensively addressed the previous concerns of the reviewers. This is a much improved manuscript. I have a few minor changes and suggestions.
Again, the manuscript shows a fluent English writing, with better technical descriptors. The literature has been improved since the last version. The article is structured as a comprehensive unit, with figures and tables and supplemental material correctly placed.

Experimental design

HGT phylogeny analysis is better formulated with more accurate descriptors.
Questions addressing data collection ((RNA extraction, mapping reads, analysis) have been addressed. Though I have a few comments:
-Line 109
The authors have included the query source, M. leidyi gene models (ML2.2). Could it be possible to have its reference, how the gene model was created? And if there’s no reference, can the authors explain how they get it (methodology)? Maybe it can be included in the supplementary material.
-Line 185
HGT developmental expression profiles
This section was clearly improve, but it has raised me a few more question:
-Line 197 “Since the CEL-Seq protocol considers only the 3’ end of transcripts”
Just a quick check on that protocol, it states that is “Strongly 3’ biased”. I’d suggest to change the manuscript sentence with the term “biased”. Question: have the authors check their reads “bias” along the transcripts where they are mapped?
-Line 198
I am confused with the term used by the authors in order to calculate their reads per million, no considering the transcript length. They use tpm (transcripts per million), but if you look in the literature, you find TPM (transcripts per million) which takes the transcript length into the equation (eg. Wagner GP, Kin K, Lynch VJ. Measurement of mRNA abundance using RNA-seq data: RPKM measure is inconsistent among samples. Theory Biosci. 2012). I would suggest to use the term and acronym reads per million (rpm); which meaning and calculation is pretty straight forward.
-Line 206
Sorry, but it’s still not clear to me how the tpm (or rpm) are calculated giving the replicas. So, after counting the reads per transcript (htseq-count),the authors calculate the tpm (or rpm) for each replica? Is that correct? And then the median tpm values (for all the replicas in each gene?) are summed? I’d appreciated if they can rephrase it to facilitate reader comprehension.

Validity of the findings

'no comment'

Additional comments

Minor suggestions
- Figure 1
The authors have included Figure 1 in order to show the pipeline in each scenario, with the number of genes included. This has help a lot in the understanding of the manuscript. But since the genes IDs are not included, the reader (at least me) still needs to go back and forth all the time to follow each step and what genes are represented. I originally suggested to implement this on a table S2, but since the authors designed Fig. 1, could it be possible to correlate Fig. 1 with table S2; like include in table S2 some coding (color, A-B-C...) to refer in which step/steps is included each gene?
- Figure 3
The boxes with “Confirmed” and “Unconfirmed” are missing, like in the previous version and in Fig S3. Please, let me know if it’s intentional or if not, it can be fixed.
-Line 66
It will be worth to include here some literature regarding false negatives/positives in BLAST searches, as addressed in the discussion (eg. Longo 2011, Merchant 2014)
-Line 270
It is not clear to me if the scaffold with geneID ML49231a has other genes at all, or the genes nearby don’t have any introns.

---

## Round 0.3 · accepted · Accept

I believe all comments were succesfully addressed. The few remaining issues, such as the use of transcripts per million instead of reads per million are of a broader discussion nature than is appropriate to continue during the review process. I consider this paper ready for publication!

Congrats,

dan

#